# Pigeon Circovirus over Three Decades of Research: Bibliometrics, Scoping Review, and Perspectives

**DOI:** 10.3390/v14071498

**Published:** 2022-07-08

**Authors:** Benji Brayan Ilagan Silva, Michael Louie R. Urzo, Jaymee R. Encabo, Alea Maurice Simbulan, Allen Jerard D. Lunaria, Susan A. Sedano, Keng-Chih Hsu, Chia-Chi Chen, Yu-Chang Tyan, Kuo-Pin Chuang

**Affiliations:** 1International Degree Program in Animal Vaccine Technology, International College, National Pingtung University of Science and Technology, Pingtung 912, Taiwan; g10985004@mail.npust.edu.tw; 2Microbiology Division, Institute of Biological Sciences, College of Arts and Sciences, University of the Philippines Los Baños, Los Baños 4031, Laguna, Philippines; mrurzo@up.edu.ph (M.L.R.U.); jrencabo@up.edu.ph (J.R.E.); asimbulan@up.edu.ph (A.M.S.); adlunaria@up.edu.ph (A.J.D.L.); 3Graduate School, University of the Philippines Los Baños, Los Baños 4031, Laguna, Philippines; 4Veterinary Vaccines Laboratory, National Institute of Molecular Biology and Biotechnology, University of the Philippines Los Baños, Los Baños 4031, Laguna, Philippines; sasedano@up.edu.ph; 5Language Center, National Pingtung University of Science and Technology, Pingtung 912, Taiwan; awilliamhsu@yahoo.com.tw (K.-C.H.); ss77ss77.tw@gmail.com (C.-C.C.); 6You Guan Yi Biotechnology Company, Kaohsiung 807, Taiwan; 7Department of Medical Imaging and Radiological Sciences, Kaohsiung Medical University, Kaohsiung 807, Taiwan; 8School of Medicine, Kaohsiung Medical University, Kaohsiung 807, Taiwan; 9Graduate Institute of Medicine, College of Medicine, Kaohsiung Medical University, Kaohsiung 807, Taiwan; 10Institute of Medical Science and Technology, National Sun Yat-Sen University, Kaohsiung 804, Taiwan; 11Department of Medical Research, Kaohsiung Medical University Hospital, Kaohsiung 807, Taiwan; 12Center for Cancer Research, Kaohsiung Medical University, Kaohsiung 807, Taiwan; 13Research Center for Environmental Medicine, Kaohsiung Medical University, Kaohsiung 807, Taiwan; 14Graduate Institute of Animal Vaccine Technology, College of Veterinary Medicine, National Pingtung University of Science and Technology, Pingtung 912, Taiwan; 15School of Dentistry, Kaohsiung Medical University, Kaohsiung 807, Taiwan; 16Companion Animal Research Center, National Pingtung University of Science and Technology, Pingtung 912, Taiwan

**Keywords:** bibliometrics, circovirus, pigeon circovirus, young pigeon disease syndrome

## Abstract

The pigeon circovirus (PiCV), first described in the literature in the early 1990s, is considered one of the most important infectious agents affecting pigeon health. Thirty years after its discovery, the current review has employed bibliometric strategies to map the entire accessible PiCV-related research corpus with the aim of understanding its present research landscape, particularly in consideration of its historical context. Subsequently, developments, current knowledge, and important updates were provided. Additionally, this review also provides a textual analysis examining the relationship between PiCV and the young pigeon disease syndrome (YPDS), as described and propagated in the literature. Our examination revealed that usages of the term ‘YPDS’ in the literature are characterizations that are diverse in range, and neither standard nor equivalent. Guided by our understanding of the PiCV research corpus, a conceptualization of PiCV diseases was also presented in this review. Proposed definitions and diagnostic criteria for PiCV subclinical infection (PiCV-SI) and PiCV systemic disease (PiCV-SD) were also provided. Lastly, knowledge gaps and open research questions relevant to future PiCV-related studies were identified and discussed.

## 1. Introduction

Published in 1992 in the “In my experience…” section of the *Journal of the Association of Avian Veterinarians* is a five-sentence report marking the earliest record on circovirus infections in pigeons. The report describes an infection that is morphologically diagnosable by examination of the presence of inclusion bodies in necropsy tissues, commonly or sometime only found in the bursa of Fabricius of the birds. It was noted that the virus is “apparently distinct from the psittacine virus” [1]. Of importance here is that neither the psittacine virus mentioned, which was later renamed as the beak and feather disease virus (BFDV), nor the family *Circoviridae* were officially recognized yet [2].

While no earlier records were recoverable, the report mentioned above also seemingly reflects what is then an on-going circulation of information among practitioners regarding the existence of and diagnostic methods for the detection of this virus. According to subsequent reports, this virus was first recognized in the late 1980s in Canada and Australia [3]. It is similarly from this report that the current review started tracing the progress in pigeon circovirus (PiCV) knowledge in the 30 years of research and studies that followed up to this day.

Aided by bibliometrics, the current review sought to map the PiCV research landscape as it expanded for the past three decades. Bibliometrics, a multidisciplinary approach crossing fields of mathematics, statistics, and philology to analyze knowledge carriers quantitatively, is commonly employed to assist in organizing records from a field of study or a specific research topic to identify historical trends and locate research hotspots and frontiers. Bibliometric tools are employed to perform a variety of analyses producing measures related to authorship, sources of records, collaborations, citations, and keywords association and co-occurrence, among others. Previously, reviews employing bibliometrics and original bibliometric studies on several viruses, including coronaviruses, oncolytic viruses, human immunodeficiency virus, norovirus, Japanese encephalitis virus, African swine fever virus, and other veterinary and human pathogens, have been published in various scientific journals [4,5,6,7,8,9,10,11,12,13].

Primarily divisible into three general goals, this review aims to (1) present a bibliometric assessment of the entire accessible PiCV-related research corpus, and in tandem with bibliometrics; (2) present a synthesis of the historical developments, current knowledge, and important updates; and (3) provide a critical analysis to identify new perspectives, knowledge gaps, and open research questions.

## 2. PiCV-Related Knowledge Production—Bibliometrics

Preliminary searches for this review were conducted on the Dimensions, PubMed, and Scopus databases. The search terms included were ‘pigeon circovirus’, ‘columbid circovirus’, ‘young pigeon disease syndrome’, and a combination of these three terms joined by the operator ‘OR’. The searches were conducted in February 2022, and all studies were included for evaluation regardless of publication time or form. Additional records were retrieved by manual searches in Google Scholar and ResearchGate. Upon retrieval of the potentially relevant articles from the databases, a unified list was generated following the Dimensions database format, and records were deduplicated. Preliminary assessments of the records based on the title and abstract were performed by four reviewers working in two independent groups. Differences were resolved by discussion after the assessment of the recoverable information of each of the records.

Articles were selected if they were at least able to demonstrate detection of pigeon circovirus (either as the main virus of interest, or as a virus concurrently investigated with other pathogens concerning a disease or condition), or if they were able to at least show an appreciable discussion of aspects directly related to PiCV research, for original and review papers, respectively. Bibliometric analyses were performed, and visualizations were created using VOSViewer version 1.6.18 [14].

### 2.1. Research Corpus: Records, Growth and Sources

The search for pigeon circovirus-related records conducted in February 2022 yielded 1648 potentially relevant records. After the creation of a unified database of all these records following the format of the Dimensions database, 1049 records were removed as duplicates, while an additional 79 entries were deemed as erroneous database entries from the sources. Therefore, 520 records were screened for eligibility based on the minimal criteria set above, wherein 389 records were determined as out-of-scope, and two others were eliminated as they only contain correction notes of relevant records that were already part of the eligible entries. A diagram of the flow of records through the steps of this review is presented in Figure 1.

Of these 129 included records, 99 (76.7%) were original articles including research publications in journals, short/brief/preliminary communications, a government agency report, a conference paper, a dissertation manuscript, and two pre-prints, while the remaining 30 (23.26%) were review papers and book chapters. A total of 117 records were retrieved in full text, while the remaining 12 were mostly either older records or in languages other than English or Chinese. Regardless, other pertinent information (abstracts, publication type, publication year, source title, authors, authors’ institutional affiliations, author’s county of affiliation, etc.) on all the 129 records was retrieved as much as we could for subsequent bibliometric analyses. 

From the earliest record that we were able to retrieve [1], PiCV-related publications consistently grew in a linear manner in the following three decades, as shown in Figure 2a provided below.

While the first decade of PiCV research saw just a couple of new records per year, remarkable is the first peak in the number of new records (Figure 2b) at the turn of the millennium that saw the first use of polymerase chain reaction for the detection and cloning, and the subsequent sequencing, alignment, and phylogenetic analyses of the PiCV genome [15,16,17,18,19], as well as the first report of pigeon circovirus detection from Africa [20] and Asia [21]. It must be noted that the record of PiCV detection from the African continent [20] was not recovered by our search strategy but was however found during the examination of the texts of the included records. This thereby highlights one of the necessities for this review because this record has also been missed in some of the most recent reviews [22,23,24].

A few succeeding years following the availability of the first whole genome sequence of PiCV seemingly looks like a lull in the history of PiCV research. However, examination of the records produced during the years 2001 to 2004 revealed a period wherein the apparent focus of research works seemed to be the utilization of the sequence information for the development and evaluation of new and more specific molecular methods for the detection of this circovirus infection [15,16,17,18,19,25,26,27,28,29]. The adaptation of these developed methods manifested in the studies that soon followed that demonstrated other potential host of the virus, its transmission, epidemiology, and organ/cellular tropism [30,31,32,33,34,35,36].

One hundred fourteen (114) of the 129 included records for this review were made available through publications in peer-reviewed academic journals, and further examination revealed that almost a third (31/114; 27.2%) of all these peer-reviewed publications were published in only three journals. The 11 journals with the greatest number of PiCV-related publications also account for 49.6% (64/129) of all included records even though there were 51 journals identified, suggesting possible authorial preference for highly specific and specialized journals, like *Avian Pathology*, *Avian Diseases*, and Journal of *Avian Medicine and Surgery*, over more general veterinary- or virology-related journals (Table 1).

### 2.2. International and Institutional Collaborations

Co-authorship analysis by country showed that there are 39 countries listed as the authors’ affiliations. Most productive among which are the United States with 27 records retrieved and included for this review and collaborating with 20 other countries, Northern Ireland with 25 records and 8 international collaborations, and Germany with 17 records and 3 collaborating countries (Figure 3a). These countries are followed by Belgium, Poland, China, Taiwan, Australia, and Italy.

On the other hand, several records were also observed to have been produced, or co-produced from countries that have not had outside collaborations (Figure 3b). Among these countries are Netherlands [37], Japan [21,38], Iraq [39], Czech Republic [29,32], and Brazil [40].

Meanwhile, ranking institutions/organizations by productivity, the Department of Agriculture and Rural Development for Northern Ireland, University of Liege, University of Warmia and Mazury in Olsztyn, Queen’s University of Belfast, University of California, Davis, National Pingtung University of Science and Technology, Leipzig University, the Agri-food and Biosciences Institute in Stormont, and the Robert Koch Institute in Berlin produced the most number of records. Additionally, both the Department of Agriculture and Rural Development for Northern Ireland, and University of Liege are both linked with 10 institutions, while University of Warmia and Mazury in Olsztyn has links with six (Figure 3b). However, when ranked by total link strength, Blood Systems Research Institute and University of California, San Francisco were the two top institutions, each with a total of 30 points. Total link strength is defined as “the total strength of the co-authorship links of a given [organization/institution] with other [organizations/institutions] (p. 5)”, whereas links are defined as “number of co-authorship links of a given [organization/institution] with other [organization/institution] [41]. These observations highlight the central role of the western countries and institutions both in terms of number of record production (in extension, knowledge production), and international institutional collaborations in the field of PiCV research, compared to countries and institutions from other regions.

### 2.3. Author Collaborations and Citations

Visualized in Figure 4 is the collaboration map among authors who worked on PiCV-related research. In total, 455 individuals were identified to have authored or co-authored the retrieved records included in the current review. In general, as observed in institutional collaboration mapping above, clusters of collaborations were identified among authors. Most notable among which is the extensive collaboration cluster of Todd, Daniel of the Agri-Food and Bioscience Institute of the Department of Agriculture, Environment and Rural Affairs (DAERA), which succeeded the Department of Agriculture and Rural Development for Northern Ireland, shown above to have the most number of institutional links (Figure 3b). Previously working on other viruses with circular genomes, particularly chicken anemia virus, psittacine beak and feather disease virus, and porcine circovirus, Todd’s first publication released in 2000 is also the most cited record retrieved in this review [42] based on the data gathered from the Dimensions database. Detailed in Table 2 are the top authors ranked by the number of retrieved records, with their respective citations, total link strength, and year of last publication.

Inspecting the details of the author collaboration networks, it is also notable that among the authors listed in Table 2, Todd is linked with Smyth, J.A., Duchatel, J.P., Raidal, S.R., and Shivaprasad, H.L., which demonstrates the tightly knitted network of the most prolific PiCV researchers (Figure 4). Important however is that this network published most of their work in the early years of PiCV research as indicated by the year of their most recent record (Table 2), except for Raidal, S.R. and Shivaprasad, H.L, who both published in 2019. While the previously mentioned cluster had links in other European countries, in two separate other networks, Mankertz, A. and Soike, D., and Muller, H., were also leading central roles in PiCV knowledge production in Germany in the first two decades of the PiCV research field. In Poland, Stenzel, T.A., Koncicki, A., and Tykalowski, B. were also working on PiCV-related research since 2012 and currently active in producing new publications. On the other hand, among the workers in Asia, only Chuang, K.-P. of Taiwan ranked within the top 10 most prolific authors (Table 2). 

Cursory assessment of the most cited records showed a higher number of citations on reviews with a wider topic, i.e., not exclusively about pigeon circovirus, and those that were published earlier. Among the most important were two reviews on avian circoviruses by Todd and co-workers in 2000 and in 2004 [42,43], and the review on important viruses of pigeons by Marlier and Vindevogel [35]. Interestingly, two original articles exclusively on PiCV were also among the most cited documents. These were the papers that reported the first cloning and sequencing of the PiCV genome [15], and the paper that effectively formalized the association between PiCV and the young pigeon disease syndrome (YPDS) [44].

## 3. PiCV Knowledge—Current Understanding and Updates

With a wider understanding of the landscape of PiCV research across the world, this review also aims to systematically synthesize the current knowledge of the field from its historical development to locate its status and the gaps that should be addressed in the future. 

### 3.1. The Virus

Pigeon circovirus belongs to the species *Pigeon circovirus*, genus *Circovirus* of the family *Circoviridae*. The species was formally recognized as a new member of genus *Circovirus* in 2005. According to the taxonomic report of the International Committee on Taxonomy of Viruses (ICTV) July 2019 release (EC 51, Berlin, Germany), the family *Circoviridae* is included in the order *Cirlivirales*, class *Asfiviricetes*, phylum *Cressdnaviricota*, kingdom *Shotokuvirae*, and realm *Monodnaviria*. No changes were made on these taxonomic assignments during the most recent report released in 2021 (ratified in 2022) [2].

Assignment to a given genus within the family *Circoviridae*, either as a member of genus *Circovirus* or *Cyclovirus*, is based on the location of the putative origin of replication *ori*. Circoviruses have the *ori* on the same strand encoding the replication-associated protein (Rep), while the *ori* is located on the capsid protein (CP)-encoding strand for the cycloviruses. An 80% genome-wide nucleotide sequence identity is set as the demarcation criteria for circovirus species based on the distribution of pairwise identities [45,46]. 

Primarily understood based on studies on porcine circoviruses, members of the family *Circoviridae* are characterized to have non-enveloped virions with icosahedral T = 1 symmetry. Pigeon circovirus virions were observed to be ranging from 14–25 nm in diameter [16,47]. Circovirus genomes are monopartite, circular, single-stranded DNA of approximately 1.7–2.1 kb in length which are replicated via rolling circle replication [46]. PiCV has approximately 2030 bp genome, relatively larger than other circoviruses [22]. A nonanucleotide motif ‘(T/n)A(G/t)TATTAC’ marking the *ori* is found at the apex of a potential stem-loop structure located at the intergenic regions between the 5’-ends of Rep- and Cp-encoding open-reading frames, which are also known as ORFs V1 and C1, respectively. It is held that the Rep initiates the rolling circle replication by creating a nick on the virion-sense strand between the 7th and 8th positions in the conserved nonanucleotide motif. Except for the chicken anemia virus genome, all circovirus genomes have an ambisense organization with the Rep encoded on the virion-sense strand, while the Cp is encoded on the complimentary strand of the replicative form dsDNA [22,46,48]. The PiCV genome also encodes three other ORFs; the functions of the corresponding proteins are still yet to be known [22,49].

### 3.2. Geographic Epidemiology, Host Range, and Genetic Diversity

Prior to having the sequence of the pigeon circovirus, detection and diagnosis of infection were heavily reliant on histopathological examinations and electron microscopy findings of intracytoplasmic and/or intranuclear inclusion bodies in lymphoreticular and hepato-intestinal tissues. There were no means of verification for viral identification, and earlier studies can only rule out the possibility that the infecting agent is the Beak and Feather Disease Virus (BFDV) by employing immunologic assays, in situ hybridization, and PCR techniques specifically targeting this previously characterized pathogen [47,50,51,52,53,54,55]. In the early 2000s, Mankertz et al. provided the first complete sequence of the PiCV genome, opening an avenue for a more accurate detection of the virus in infected pigeons [15]. Following its publication, development and evaluation of nucleic acid-based detection methods, such as PiCV-specific PCR techniques, in situ hybridization, and dot blot analysis, soon followed [18,26,27,31,34].

Later on, detection methods coupled with quantification strategies were developed into real-time quantitative and digital droplet PCR assays with the aim of correlating viral load to the clinical status of the host [56,57,58]. Rapid and accurate detection was also developed employing loop-mediated isothermal amplification that was reported to have no cross-reactivity against porcine circovirus 2 (PCV2), which is unlike the in situ hybridization technique that not only detects PCV2 but also the more closely related BFDV [19,59]. High-throughput sequencing is by far the most advanced technology ever used that detects PiCV, but the complexity, cost, and duration of result turn-out make this method unsuitable for diagnostic applications [60].

*Columba livia domestica*, either kept and bred for different purposes (sporting/racing, fancy, or utility) or feral, is known for its cosmopolitan distribution [61], and so are PiCV infections [23]. Considering that racing, the most common reason for pigeon breeding and raising [61,62,63,64], allows for the close interaction of pigeons from different local and even international lofts during transport and the actual competition, it is not surprising therefore that pigeon circovirus has also been detected in many countries across the world. Sporting events also provide opportunities for intermingling among feral and racing pigeons. Additionally, international trade for pigeons with good pedigree to be used as breeding stocks has also been suggested to facilitate the international distribution of this virus [22,65,66,67].

Retrieved records for this review showed that the first descriptions of case examinations demonstrating the presence of circovirus infections in pigeons were first made available in 1993 from studies in the United States and South Africa [20,50,68]. Although, based on other accounts, the virus has also been observed as early as the late 1980s in Canada and in Australia [3,54].

Within the first decade after the short report of Schmidt [1], retrieved records showed detection of the virus in several other countries including Northern Ireland [16,34,69,70], Germany [15,44], Italy [31,71], France [72], Belgium [30,33,34], and Japan [21,38]. Meanwhile, between 2003 and 2012, researchers in Czech Republic [29,32], Poland [36,58,67,73,74,75,76,77,78,79,80], Denmark [30], Slovenia [81,82], Hungary [66,83], United Arab Emirates [84], Libya [85], Nigeria [83,86], and Taiwan [59,87,88,89,90,91,92,93] reported first local detections of PiCV. Lastly, during the most recent decade of PiCV research, new publications and reports were also produced from Brazil [40], Iran [94,95], Iraq [39], and Turkey [96]. Other than the first local detection report for each country indicated above, succeeding reports demonstrating detections of PiCV were also cited to add to, if not complete, previous reviews.

Additionally, while no literature records were retrievable from Croatia and Algeria, sequences of PiCV detected from these two countries in 2010 (KP773230.1) and 2017 (MH932546.1), respectively, were available in the National Center for Biotechnology Information (NCBI) database. Shown in Figure 5 is a world map indicating countries with reported detection of PiCV colored according to the earliest retrieved record of detection report.

Attempts to identify infecting circoviruses in other avian species have also uncovered evidence of PiCV cross-species infection. Relying only on serological assays using hemagglutination targeting BFDV in wild Senegal doves (*Streptopelia senegalensis syn. Spilopelia senegalensis*) presenting similar symptoms to psittacine beak feather disease has proven that the infecting agent is a different circovirus [52]. In a retrospective study later on, PiCV infection was confirmed in *S. senegalensis* after performing sequence comparison [49]. Relying on a combination of clinico-histopathological observations and probe hybridization techniques, other earlier studies on circovirus-like infections in other bird species also drew suspicion of PiCV infection. However, unlike the case of PiCV infection in Senegal doves, these other cases were later proven to be caused by entirely new circovirus species [42,53,97,98].

A shift in diagnostics occurred when the PiCV whole genome sequence was made available, allowing sequence comparison as a form of verification [15]. This was proven to be useful when PiCV was detected in Eurasian collared dove (*Streptopelia decaocto*) using the nested PCR approach targeting the partial capsid gene of the virus [32]. Meanwhile, degenerate primers targeting the conserved region of the *rep* gene in circoviruses was amplified in tissue samples from chicken obtained from Nigeria [86]. 

Moreover, a recent study involving outbreak of Chlamydia psittaci in aviaries in Taiwan surprisingly detected PiCV in 12 species of birds (*Aix galericulata*, *Cygnus melancoryphus*, *Caloenas nicobarica*, *Columba pulchricollis*, *Goura cristata*, *Ocyphaps lophotes*, *Spilopelia chinensis*, *Streptopelia orientalis*, *Pavo cristatus*, *Threskiornis aethiopicus*, *Phoenicopterus chilensis*, *Phoenicopterus ruber*) coming from five different orders by amplifying the partial cap gene of the virus. The authors have correlated the lethal outbreak of chlamydiosis to a concurrent circulation of PiCV which may have “play[ed] a key role in augmenting [the] disease progression” although further proof certainly is needed [92]. Moreover, PiCV detection is not only limited to different species of birds as it was also detected in ticks parasitizing camels and sheep from Inner Mongolia. The observation was published as a preprint. It is notable though that the whole genome sequence of PiCV (Accession Number: MN920392) detected from these ticks was recovered using high throughput and Sanger sequencing verified by PCR [60]. 

From these studies, partial or complete PiCV sequences were obtained from 15 avian species and one from a parasitic arachnid (Figure 6).

Although reportedly first observed in Canada and Australia, the origin and duration of the circulation of PiCV are unknown. The first whole genome sequence provided a reference point for its taxonomic classification. In relation to BFDV, PCV1, and PCV2, genomic sequence homologies between these viruses and PiCV were reported to be 57%, 36%, and 34%, respectively [15,16]. 

Using the 371 available PiCV whole genome sequences retrieved from NCBI, a pairwise identity calculation showed that the two sequences with the lowest identity similarity (83.201%) are sequences from Japan (Accession Number LC035390) and China (Accession Number KJ704801), which is consistent with the study of Khalifeh et al. (2021) that reported >83% genome-wide identity [73] (Appendix A). The same study reported as many as 95 PiCV genotypes based on a 98% pairwise identity genotype demarcation criterion [73]. This is also consistent with our observation based on the previously mentioned 371 whole genome sequences. Shown in Appendix A are the resulting phylogenetic trees from the same dataset. Topology and grouping of sequences widely reflects that of a previous report [73]. 

Most of the variability, according to earlier studies, is attributable to differences in the *cp* gene sequence [22,49,67,74,99,100,101]. Between the *rep* gene and the *cp* gene, sequence analyses have shown that the *cp* sequence undergoes more frequent mutations as an adaptive response to the hosts’ immune system driving the evolution of circoviruses primarily by positive selection, although negative selection has also been observed. As a whole, these mutations in the *cp* explains the high genetic variability among PiCV sequences [22,49,66,67,75,88,99,100]. One the other hand, potential genetic admixture with other circoviruses has also been observed in Australia [102].

More recently, wider interest and more studies focused on the role of recombination as an evolutionary driver of circovirus evolution have been published [40,66,73,76,102]. Besides the observed frequent mutations in the *cp* gene, this observed high genetic diversity in PiCV sequences is also attributed as a product of constant recombination driving the evolution of this virus [75,88]. To understand the genetic recombination in PiCV, in an experimental study, infected and non-infected pigeons were housed in a single loft for a period of 23 days to observe recombination events [73]. Recombination detection analysis of the 178 whole genome sequences from this study revealed 13 recombination events, with event 2 as most common recombination event present in 109 genomes. Recombinant event 2 was shown to be located at the 5’ portion of the *rep* spanning a ~100 nt region. Additionally, localizations of the recombination hotspots were also detected at the 3’ region of *rep* and the intergenic region, while a cold spot was identified near the 5’ region of the *cp* [73]. The authors note that these observations were different from previously detected hotspots, which were located near the 3’ end of the *rep* and *cp.* It was also reasoned that the detected recombination cold spot located in the *cp* might be due to importance of the capsid protein to the transmission and infection ability of the virus.

Similarly, using the 371 whole genome sequences that we were able to retrieve, we also attempted to perform a recombination detection analysis of this global dataset. The analysis was performed as in [73] using RDP5 [103]. Our preliminary analysis revealed 130 detected hypothesized recombination events, of which 105 were accepted based on the detection of these events by at least three methods employed by the program. In our analysis, localizations of the recombination hot spots were observed at the 5’ regions of the *rep* and *cp*, as well as the 3’ region of *rep* up to the intergenic region, while recombination cold spots were observed at the central region of *cp* (Appendix A). These observations seemingly merges together previously differing detection of recombinant hot spots, while at the same time consistent with previously reported observations of cold spots [73,75]. After removing the accepted recombinant regions, shown in Appendix A is the resulting phylogenetic tree. We note that the grouping of sequences into clades remain largely the same relative to Appendix A, but the positions of the clades within the tree differ, particularly the relationships of genotypes 5 and 6 to genotype 4, and genotype 7 to genotype 4. To explain this, we suggest that an assessment of each of the recombination events among these sequences be evaluated in more detail.

Up to this point, efforts to group PiCV sequences yielded varying results. Initially, two main clades were reported, however subsequent reports described phylogenetic reconstructions with as many as nine clades [38,40,49,67,101,104]. Despite multiple attempts in classifying PiCV into groups, there was no observed association between clades and geographic origin or the pathogenicity of the virus [22,49,73,100]. We however note that from the trees generated in this review, there tends to be clustering of sequences from individual studies, which might suggest association of PiCV sequence grouping to temporality, or geography or location of isolation, or an interaction of both. However, it must also be emphasized that while the current analysis utilized all available whole genome sequences, populational representation (or the lack thereof) of the “true” circulating strains might have also contributed to current observations. Of the 371 genomes used in this review, around 200 were from Western countries, particularly from Poland, which has contributed ~160 whole genome sequences that are mostly from one single study [73]. More samples of PiCV whole genomes are encouraged to be sequenced, especially from underrepresented regions and/or countries to come up with a better resolution of this observed and/or hypothesized tendencies/associations.

### 3.3. Transmission, Infection, and Control

Pigeon circovirus can be transmitted through both horizontal and vertical pathways. Detection of the virus in the intestine, cloaca, and faeces supports faecal–oral route transmission [18,33,50]. Although, inhalation of other faecal contaminated materials, such as feather dust, has also been suggested as a potential respiratory route of infection. The virus has previously been reported to be detected in the pharynx, trachea, and lungs, suggesting that respiratory tissues may be a major potential site for virus replication and persistence, especially in older pigeons [33]. On the other hand, vertical transmission was proven by the detection of the virus in testis and semen samples of breeding cocks, the ovary but not in the oviduct of the hens, in embryonated eggs, and chicks recovered from eggs shortly before hatching [18,30,33,56].

Detection of the virus in pigeon embryos was observed to be more common than previously thought. Reports have shown that about 11–36% of embryos may have already been infected with the virus before hatching [30,33]. No specific embryonic tissue seemed to be targeted by the virus infection as DNA of the virus has been detected in pharynx, trachea, lung, liver, spleen, intestine, kidney, heart, and bursa of Fabricius. Interestingly, viral DNA is not detected in embryonic bone marrow [33]. Observation of the status of viral infection during the first few weeks from hatching showed that while detection of the virus DNA in cloacal swabs of pre-weaning (up to 28 day old) chicks were only ranging from 1–20%, weaning and the subsequent transfer to the rearing loft coincides with the detection of the virus in 100% of the birds, suggesting high probability of virus transmission and new infection in rearing lofts [33]. In addition, noteworthy is the observation that although viral DNA was observed in crop tissues of young birds by in situ hybridization, detection of the viral DNA by PCR of crop swab samples yielded no positive result, suggesting that excretion and transmission of the virus through the crop milk is rare [30].

Pigeon circovirus infection is commonly investigated and reported in pigeons less than 1 years of age, most of which were not exhibiting any clinical sign albeit the high viral loads observed in multiple tissue samples, particularly in the bursa of Fabricius [19,27,30]. Specifically, the virus has also been detected in varying percentage in different reports using different methods in tissue samples from the spleen, liver, thymus, kidney, crop, intestine, brain, trachea, lung, heart, blood, bone marrow, esophagus, Peyer’s patch, nose, and the third eyelid [19,27,30,89]. Meanwhile, among older pigeons (more than 1 year old), detection of the virus is more common in spleen, and respiratory tissue samples than in any other organs. Compared to younger pigeons, detection of the virus in the blood, pharyngeal and cloacal swabs of older pigeons is lower, while detection in other organs remains high. Taken together, these observations suggest recovery of these older pigeons from the infection. However, this also suggests that total elimination of infected pigeons from the breeding stock would not be possible [30,33].

Demonstration of the capacity of PiCV to cause clinical disease manifestation had been challenging due to both the absence of a reliable method to propagate this virus and specific pathogen free-certified pigeons. However, purified virus particles from infected tissues have been used to experimentally challenge pre-screened pigeons. The challenged birds did not display clinical disease manifestation, however, the characteristic lesions in the bursa of Fabricius and the spleen were evident in the challenged but not in the control pigeons. PiCV DNA was also detected in cloacal swab and tissue samples of the inoculated birds. These observations support the hypothesized immunosuppressive role of PiCV in pigeons [105]. More recently, examination of the lymphocyte populations among symptomatic and asymptomatic PiCV-positive, and PiCV-negative pigeons confirmed this immunosuppressive role of PiCV. It was observed that PiCV infection induces B lymphocyte apoptosis thereby suppressing humoral immunity [58]. These two studies are perhaps the best evidence to settle contentions on the immunosuppressive role of PiCV.

To date, no commercial vaccine is available to control PiCV infections. Experimental studies on the production of the recombinant capsid protein and the evaluation of its potential use as an antigen for vaccine formulations have been reported in the literature [79,93]. Moreover, probiotics and interferons have also been reportedly explored as alternative and/or supplemental modalities to control the effects of PiCV infections [90,91]. At the moment, protection against potential detrimental effects of PiCV infection relies on good biosecurity measures in the loft. However, while biosecurity principles can be easily adapted in meat pigeon farms, the same is not true for racing pigeon lofts. Importing breeding stocks, flight training, and, importantly, race events all violate biosecurity principles [22].

### 3.4. Revisiting Young ‘Pigeon Disease Syndrome’

Like in many historical cases of new virus discovery, the emergence of a new condition negatively affecting the health of, in this case, pigeons, led to the discovery of PiCV. Retrieved records for this review show that Woods et al. (1993) provided the first case report of a circovirus-like infection in a pigeon from a flock that experienced 100% squab mortality three to four days following observation of anorexia and lethargy. The subject of the case report, a female racing pigeon, is described as “emaciated, with pectoral muscle atrophy [50]”. The histopathologic findings of this case remain as one of the most detailed of the succeeding cases reported in literature. The report also noted distinctness of this circovirus from PBFD based on the results of immunohistochemistry, DNA in situ hybridization, and polymerase chain reaction with DNA dot-blot hybridization. The findings contained in this report were echoed by subsequent reports, often with striking similarities but with unique and appreciable differences. 

For the several years that followed, reports of this circovirus infection centered on the similarities of the cases; thus, distilling unifying features observed. These features are the observation of: (1) intracytoplasmic/intranuclear inclusion bodies common in the bursa of Fabricius and the spleen, but are also sometimes observed in other tissues, that are (2) composed of paracrystalline arrays of tightly packed, nonenveloped icosahedral virions, and accompanied by (3) lymphoid depletion, evidenced, primarily, by bursal atrophy. Consistently, PiCV reports also indicated non-specific clinical signs of ill thrift, lethargy, anorexia, poor racing performance, and diarrhea. These features were often observed in conjunction with coinfections of other bacterial, fungal, and/or viral agents. These observations led to the hypothesized immunosuppressive function of PiCV that predisposes the pigeons to other infections. See [3,37,43,50,54,65,72]. Of exception to this is the paper by Tavernier et al. (2000) that investigated pigeons that were presented for necropsy after dying spontaneously or suffering from clinical disease. The report noted that the proportion of examined cases with concurrent infections and the nature of these coinfections, lesions, and clinical signs were similar among pigeons whether or not circovirus infection was detected, but the mortality is more common among circovirus infected pigeons. The paper added that clinical indications of the immunosuppressive effects of circovirus were not found in their examinations, but experimental infection and additional immunological study must be conducted to verify the contending reports [55]. Challenge and immunological studies conducted years later confirmed immunosuppression [58,105].

Curiously, in 2005, the term ‘young pigeon disease syndrome’ or ‘YPDS’ first appeared in the literature we managed to retrieve in full-text [44]. The paper also considered what was then called “swollen gut syndrome” as an equivalent designation, also adding that other non-infectious and infectious causes, “including *Spironucleus columbae*, *Escherichia coli* and avian adenoviruses, have been considered to contribute to the pathogenesis of YPDS (Dorrestein et al., 1992; De Herdt et al., 1994; Warzecha, 2002)” [44]. This epidemiological study obtained, among other information, occurrence of YPDS and the observed clinical signs [44]. Notably, the survey showed that “pigeons affected by YPDS were mainly between 4- and 12-weeks post-weaning. The clinical signs were not specific and included anorexia, depression, ruffled feathers, vomiting, diarrhoea, polyuria, and fluid-filled crop” [44].

Additionally, the abovementioned paper, also noted that *Escherichia coli* and *Klebsiella pneumoniae* were isolated more frequently from YPDS-affected pigeons, and that depletion of the splenic and bursal lymphocytes was only observed in pigeons with YPDS. All investigated pigeons with YPDS in this report were also positive for PiCV infection in at least one organ, most importantly in the bursa of Fabricius with 100% (45/45) positive detection of PiCV DNA by PCR. The report concluded that “YPDS is a multifactorial disease in which a pigeon might be a crucial factor, possibly by inducing immunosuppression in infected birds” [44]. This paper is pivotal since it provided a de facto definition of YPDS that would be cited in almost all subsequent reports.

## 4. Conceptualizing PiCV Diseases (PiCVDs)

### 4.1. Perspective—Problematizing ‘Young Pigeon Disease Syndrome’

Prior to the 2005 epidemiologic report [44] that popularized the use of the term ‘young pigeon disease syndrome’ or YPDS, we note that the manner of reporting and the description of observations flow unidirectionally; that is, by our reading, the provided clinical picture of having non-specific clinical signs plus the three main unifying features noted above were describing PiCV infection. We also note that we did not encounter any specific terminology that referred to a conceptualization of the abovementioned set of conditions as either a disease (multifactorial or otherwise), a syndrome, a disease syndrome, or a disease complex. If any, the disease *is* PiCV infection, or, more specifically, an acquired immunosuppression caused by PiCV infection resulting to predisposition to secondary infections. Afterall, these reports were exclusively describing cases of PiCV infection. See [3,37,43,50,54,65,72].

In our attempt to trace the term ‘young pigeon disease syndrome (YPDS)’ prior to [44], we tried to retrieve the records cited (i.e., “Dorrestein et al., 1992; De Herdt et al., 1994; Warzecha, 2002”) that, according to the paper, considered the contribution of other infectious agents in the pathogenesis of YPDS. It is unfortunate that we were not successful. It was indicated in the reference list that the first two were conference proceedings, while the latter was an article in a magazine specializing in racing/carrier pigeons [44]. However, in another published work by De Herdt and co-workers published in 1995 reporting on an epidemic of fatal hepatic necrosis, PiCV and its association with immunodeficiency were also mentioned and the paper also cited the same paper by Dorrestein et al. (1992), but never positively identifying a condition called ‘young pigeon disease syndrome’ or any specific terminology that bared similarity in construction [51]. The same is true with the report co-authored by De Herdt published in 2000 [55]. Considering these observations, our sense is that ‘young pigeon disease syndrome (YPDS)’ in this exact form or, probably, in other translational equivalents, as a coined term (as opposed to a describing phrase) were not mentioned in the cited texts, but the clinical picture of the conditions described therein and that of YPDS were similar.

With the appearance of the term YPDS in the PiCV literature [44], an apparent reversal of the flow of attribution was also observed. While reports prior to 2005 were describing observed clinical presentations and the accompanying histopathological observations of PiCV infected birds, the epidemiologic report [44] associated what was first observed as a set of clinical presentations (termed YPDS) to PiCV infection. Nevertheless, the term ‘young pigeon disease syndrome (YPDS)’ seems to be widely used by 2005, such that the same report was able to conduct an epidemiological survey on YPDS using a questionnaire administered to German breeders of racing pigeons. These, to us, suggesting that the term “young pigeon disease syndrome (YPDS)” might have originated in a lay characterization among fanciers of an observed illness or set of clinical manifestations that gained popular currency before it was eventually adapted by veterinary health practitioners and researchers, and reported in the scientific literature. A paper published later on supports this hypothesis stating, “In racing pigeons this combination of clinical signs (i.e., lethargy, weight loss, respiratory distress, diarrhoea and poor racing performance) is commonly described by pigeon fanciers as ’young pigeon sickness,’’’ [33].

As previously mentioned, this paper [44] became pivotal for its influence in the operational and functional use of the term YPDS, and the understanding and subsequent research on PiCV in the succeeding years.

We would like to note two key points about the paper [44] at this point:i.The phrase “might be a crucial factor” in the concluding statement, nuanced to the limitations of their methodology, did not set PiCV infection as a precondition/requirement to YPDS. As such, it logically follows that YPDS as, in the words of the authors, “a multifactorial disease” might also be possible without PiCV infection.ii.The concluding statement did not categorically identify its use of the term YPDS as either referring to the set of clinical, histopathological, and molecular observations reported, or the colloquial use of the term which only involved, in their words, “clinical signs [that] were not specific.” Hence, to our reading of the text, there exist virtually two conceptualizations of YPDS: one that is colloquial, and the other that is more technical in its requirements.

We contend that these two points are crucial to the developments in PiCV research that followed the abovementioned report, and to the studies that are still to be conducted in the future. As such, we also investigated how this paper and, in particular, the two key points noted above, were referred to, and propagated in the literature. Our underpinning questions are: (1) How is PiCV described to be related to YPDS?; and (2) What is ‘young pigeon disease syndrome (YPDS)’ as operationalized in the literature?

Examination of the retrieved texts showed that the relationship of PiCV infection to YPDS has been described in variable terms of associatedness, importance, and/or cruciality. Specifically, the role of PiCV infection to YPDS has been described in some papers as remaining “uncertain” and that it “has not yet been conclusively clarified”, while also described as “speculated as crucial”, “might be crucial”, “crucial”, “probably associated”, “associated”, “strongly associated”, “important”, “central”, “one of the putative factors”, “a key factor”, “one of the causative agents”, “among the major causative agents”, “the likely aetiological agent”, and “likely causal agent” in other records. Likewise, other texts also described PiCV infection to be “caus[al of] YPDS”, or “responsible for YPDS.” While these descriptors are not mutually exclusive of each other, these are also not equivalents. Additionally, the mechanism by which PiCV infection contributes to the development of YPDS has also been described in various descriptions of certainties, ranging from “assumed to induce immunosuppression”, “possibly by immunosuppression”, to “causes profound immunosuppression”. See [22,23,33,40,44,48,58,59,66,67,75,77,79,84,88,89,90,93,94,95,100,101,102,104,106,107,108,109,110,111,112,113,114,115,116,117]. To us, these degrees of variability among these descriptors do not seem to correlate with time. In other words, these variable descriptors, and the timing of their usages in the literature do not suggest increasing or decreasing acceptance of the importance of PiCV infection to YPDS, or the perception of certainty of immunosuppression as the mechanism for the PiCV association with YPDS. 

Unsurprisingly, we also found that the descriptions, definitions, and/or operationalized usage of the term “young pigeon disease syndrome (YPDS)” also vary, corresponding to the above noted colloquial and technical conceptualizations. Indeed, in most texts, YPDS is referred to as a “multifactorial” or “complex disease” characterized by a combination of symptoms (often described as non-specific), with no to variable level of attributed importance to histopathological markers. Consistent with this is our observation that YPDS is used to describe cases (or loft histories) prior to the detection of PiCV infection in numerous reports. In others, YPDS is a category of viral infections. These are all also while other records explicitly state that YPDS is a consequence of PiCV infection and subsequent immunosuppression, or categorically require specific conditions for YPDS diagnosis, such as the observation of both clinical symptoms and PiCV-specific histological lesions, presence of viral particles, and a demonstration of the presence of a large amount of viral DNA. See [22,23,33,40,56,57,58,59,67,77,79,80,90,93,96,100,105,109,110,111,113,117]. 

Problematizing PiCV infection and its pathology, YPDS, and their correlation with each other is not new to this review. These have been, in fact, subjects of several discussions and opinions before. For instance, Dr. Henk de Weerd pointed out, “As you know, about 10 years ago ‘everyone’ (except me) was extremely alarmed about the circovirus. According to many, that was the cause of ‘just about everything’ in terms of pigeon outbreaks, including ‘adeno-coli’,” which, still according to Dr. de Weerd, is internationally known as the “Young Bird disease’’ [118]. Similarly, in another perspective piece, Dr. Colin Walker expressed that, ‘young bird disease’ clusters numerous diseases with similar symptoms together such that “[f]anciers run the risk of labelling any young pigeon with these symptoms simply as having ‘young bird disease’ when, in fact, all they are acknowledging is that the young pigeon is sick with wasting and diarrhoea” [119]. Indeed, many other agents were considered to be associated with YPDS, including fowl adenovirus, pigeon herpes virus, pigeon adenovirus, paramyxovirus, *Chlamydia*, *Escherichia coli*, *Salmonella*, and other parasites in varying combinations with varying degrees of support [74,96,108,115,120]. Dr. Walker also opined, “the term ‘young bird disease’ is a poor one and one that I think should be abandoned” [119].

Most recently, experimental infection with the newly discovered and isolated pigeon Rotavirus A belonging to a novel clade also produced clinical manifestations that were also described as YPDS-like [117,121]. Some workers in the field expressed that the discovery of pigeon Rotavirus A and the demonstration of its capacity to cause a disease that is like YPDS seem to depreciate the role of PiCV in the etiology of YPDS [58].

Therefore, the questions of what YPDS is, what its relation to PiCV infection is, and how this term has been used and propagated in the literature, which reflects how veterinary health practitioners and researchers perceive YPDS, need critical assessment. The current review revealed that usages of the term YPDS in the literature are neither standard nor equivalent. YPDS in the literature is a diverse range of characterizations.

In consideration of the complex history of YPDS and PiCV infection, our current opinion is that YPDS, as colloquially conceptualized pointed above—that is, a combination of non-specific clinical symptoms, may still be useful as a putative diagnosis pending identification of the causative and/or confounding agents. We are also under the impression that PiCV must be categorically identified as a causative agent of the diseases that we propose below to be identified as PiCV subclinical infection (PiCV-SI) and PiCV systemic disease (PiCV-SD).

### 4.2. Proposal—PiCV Diseases Case Definition

From the early days of PiCV research, this virus has been consistently compared and likened to other related pathogens, most remarkably to porcine circovirus (PCV) and the associated disease formerly referred to as post-weaning multisystemic wasting syndrome (PMWS), in terms of basic biology, to genetic diversity, pathobiology, and approaches to treatment and control [23,42,57,79,88,90,93,99]. While the first case of PMWS was first recognized in 1991, years later than the observation of clinical cases in pigeons which may be categorized as YPDS, swine health practitioners were able to come up with diagnostic criteria fairly early. Notably, this was after much controversy and debate. Regardless, as early as 1999, three major criteria were set to establish diagnosis of PWMS, which was later renamed as PCV2-systemic disease (PCV2-SD), one of the porcine circovirus diseases (PCVDs) [122,123,124]. These requirements are “(1) presence of compatible clinical signs, mainly wasting, (2) observation of moderate-to-severe histological lesions in lymphoid tissues (lymphocytic infiltration and histiocytic infiltration) and (3) detection of moderate to high amount of PCV-2 within such lesions” [124]. Such strict definition of a case was considered acceptable by the veterinary and scientific communities worldwide considering a general reluctance to accept that PCV2 was truly pathogenic. This is in sharp contrast to the discovery of PCV3 and the acceptance of its causality in associated diseases, which was accepted much faster probably due to the previous experience with PCV2 [124]. The case of YPDS and its relationship with PiCV infection has not been similar, which as pointed above still necessitates a cohesive conceptualization and associated terminologies. 

Additional parallels between PiCV infection and PCV2 infection can be long, but we would like to highlight several important points of similarities related to pathology, and disease. Like PiCV, early experimental studies with PCV2 sought to demonstrate the capacity of the virus to produce clinical signs compatible to PMWS. In PCV2, successes were achieved mostly after inoculation with another infectious or non-infectious agent. Together with other field observations, these studies lead to the conclusion that PCV2 is immunosuppressive and a “necessary but not sufficient factor to develop the clinical disease” [122,123]. While no artificial/controlled co-infection studies on PiCV and other suspected pathogens had been reported to date, there is preponderance of evidence from case studies of PiCV infection and YPDS to demonstrate that PiCV infected birds with clinical symptoms were coinfected with other infectious agents. Additionally, analogously applied, PiCV infection cases displaying clinical symptoms fully satisfy even the strict definition of PCV2-SD. Important to point out here is the difference in the characterization of YPDS and PCV2-SD. While PCV2-SD explicitly necessitates PCV2 and its detection in associated lesions, YPDS, in most cases of usage, does not require or expresses no same certainty to the importance of PiCV. This is except in one paper by Duchatel and Szeleszczuk (2011), which, as noted in the previous section, requires three major requirements for YPDS diagnosis [23]. Remarkably, this paper has previously been cited by other workers, but rarely about its specification of YPDS diagnostic criteria. In one recent review, this diagnostic criteria was cited, but what Duchatel and Szeleszczuk categorically defines as YPDS, the review identifies as circovirosis [22,23].

From this perspective of the current understanding of PiCV and YPDS research, and guided by the lessons from PCV2 and PCVD, we draw our proposed PiCV diseases, as shown in Table 3.

Based on epidemiologic and case studies conducted in multiple countries across the world, PiCV infection is quite common among both domestic and feral pigeons, with some reports identifying that most young pigeons in an affected loft would have been infected before reaching 1 year old. Previously performed experimental infections of pigeons with PiCV infected tissue homogenates, as well as documented naturally infected pigeons without clinical symptoms, confirm that while no clinical manifestations were observed, characteristic histopathologic lesions were present in challenged pigeons [79,93,105]. Therefore, PiCV-subclinical infection (PiCV-SI) is a common disease. 

On other hand, PiCV-SD, like PCV2-SD, is a multifactorial disease in which PiCV is the strictly required factor. Like PCVDs, as PiCV has also been noted for its wide distribution similar to PCV2, cases of PiCV-SD must be considered to be of complex causality. This proposal, therefore, functionally reclassifies Duchatel’s and Szeleszczuk’s conceptualization of YPDS as PiCV-SD. As such, the previously reviewed clinico-pathological picture of PiCV infection with clinical manifestations and characteristic histological lesions would also be descriptive of PiCV-SD [22,23]. In this sense, many cases previously identified as YPDS can be classified as cases of PiCV-SD. In contrast, the proportion of the diagnosed YPDS cases that are not PiCV-SD cases cannot be assessed, due to, as pointed out earlier, variable characterizations of YPDS.

Furthermore, the current proposal is not to deny the involvement of other infectious agents in PiCV-SD. For instance, consistent with Schmidt et al. (2020), the discovery of pigeon Rotavirus A does not contradict to its potential importance in the development of what we are currently proposing as PiCV-SD. What this proposal is trying to resolve and formalize is the importance of and necessity for PiCV infection to pigeon health and the development of a particular disease. In extension, the proposal recognizes the pathogenicity of PiCV, perhaps, best evidenced by the demonstration of the virus’ apoptotic ability to B lymphocytes confirming its immunosuppressive function [52,92].

Guided by our examination of the literature, we are of the view that this proposal is the best step moving forward as an attempt to lodge a more cohesive approach to dealing with matters important to pigeon health management, and in extension, to scientific research.

## 5. Research Gaps 

Perhaps, among the most important research gaps in PiCV research is still the absence of a reliable and replicated method for isolating this viral agent. We, however, note two records reporting propagation of PiCV [96,112] achieved by inoculating PiCV-positive tissue homogenates into specific pathogen free (SPF) embryonated chicken eggs. 

In Sahindokuyucu et al. (2022), inoculation of tissue homogenates positive for PiAdV-A and PiCV into the chorioallantoic cavity and yolk sack of the SPF embryonated eggs yielded PiCV-positive allantoic fluid at the fifth passage. None of the other tested viruses were positive by PCR. Additionally, cultivation of the virus from the same tissue homogenate using primary chicken embryo fibroblast also yielded PiCV- and PiADV-A-positive supernatant. It was, however, unclear how many cell culture passages were performed [96]. Meanwhile, in Van Borm (2013), SPF embryonated eggs were inoculated in the allantoic cavity. Notably, the goal of their investigation is to determine the genetic diversity of pigeon paramyxovirus type 1 (PPMV-1) in their collection using next generation sequencing approach. However, their sequence data did not only provide information on their PPMV-1 isolates, but also revealed contaminating PiCV. RT-PCR analysis of their virus stocks, even those that were passaged in embryonated chicken eggs, were positive for PiCV [112]. Both these reports provide hope for possible isolation, purification, and cultivation of PiCV isolates. Replicating these successes would be essential as this would open various opportunities to investigate this virus in a more controlled and more detailed manner.

Particularly, the establishment of a reliable isolation protocol for PiCV would enable future research on the more basic understanding of the biology of this virus, including its specific mechanisms for entry, replication, and release, as well as the functions of the other proteins encoded its genome. In particular, the role of ORF C3 protein in PiCV is still unclear. In PCV2 and duck circovirus, ORF C3 has been confirmed to induce lymphocyte depletion by apoptosis [15,58,125,126]. Additional understanding of these basic facets may be critical to the understanding of the viral pathogenicity, and to the development of new approaches for treatment and control. Furthermore, propagation of pure isolates of this virus would also enable studies related to pathogenesis, virus–host interactions, especially during coinfections, host immunity, and recovery, among others.

On the other hand, with the reported detection of PiCV in other animals, particularly in ticks, it is tempting to speculate whether all or some of these species are hosts (definitive or intermediate) or a vector of PiCV. With most of these reports relying on partial sequences, it remains to be shown if these detected infections are of a different subtype or strain, or perhaps a completely new circovirus species that is a very close relative sharing a common origin with PiCV. Nevertheless, these reports seem to be a prelude to studies needed to uncover what might be a cross-species jumping in play in PiCV evolution, together with its implication to PiCV epidemiology, pathobiology, and genetic diversity. 

More than thirty years after the discovery of PiCV, there is still a lot to learn about this viral agent. 

## Figures and Tables

**Figure 1 viruses-14-01498-f001:**
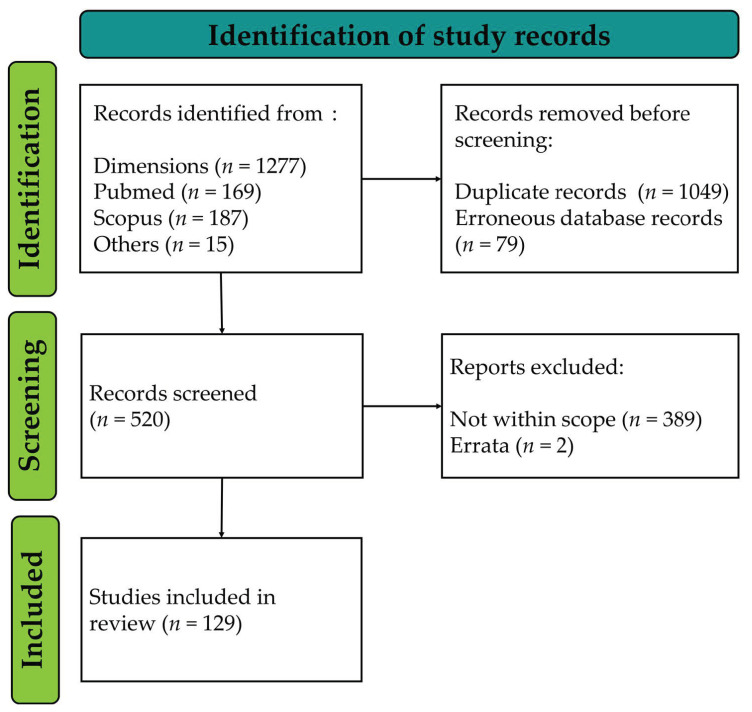
PiCV-related records retrieved from different sources.

**Figure 2 viruses-14-01498-f002:**
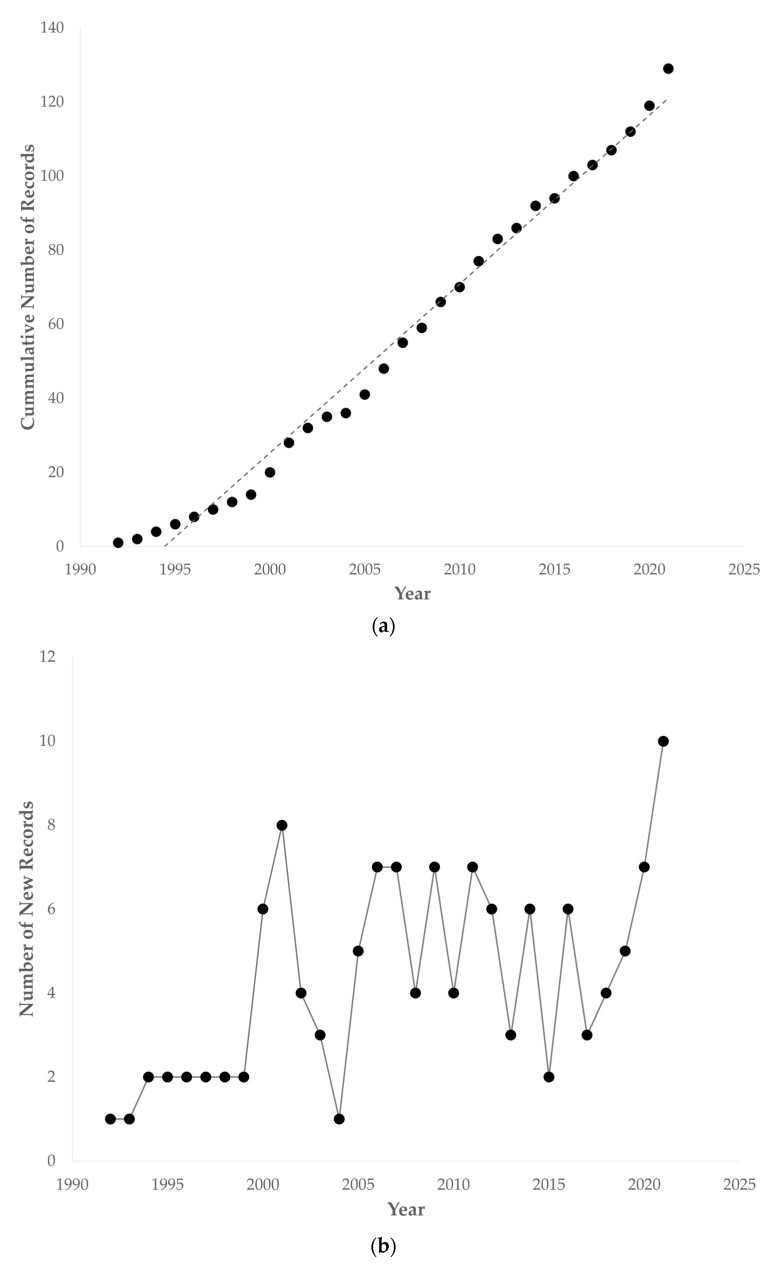
PiCV-related records retrieved from different sources shown as (**a**) cumulative number of records over the years, and as (**b**) new records per year.

**Figure 3 viruses-14-01498-f003:**
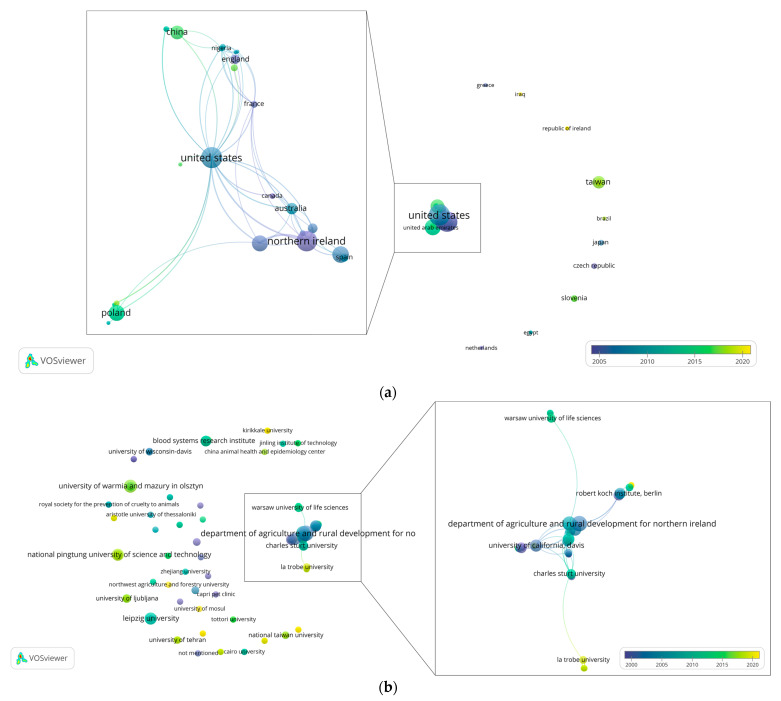
PiCV research collaboration maps among (**a**) countries and (**b**) institutions scaled by number of documents produced and colored by average publication year.

**Figure 4 viruses-14-01498-f004:**
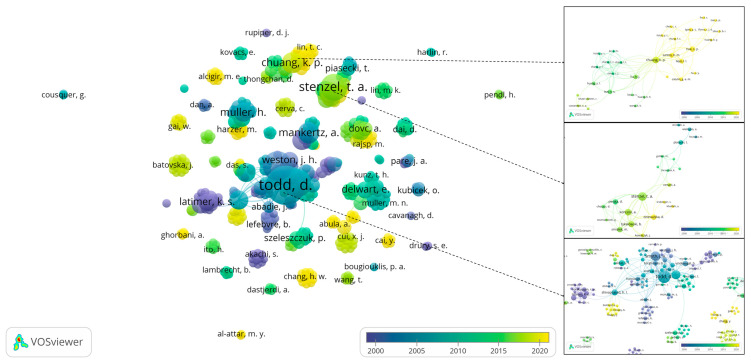
PiCV research collaboration map among authors scaled by number of documents produced and colored by average publication year.

**Figure 5 viruses-14-01498-f005:**
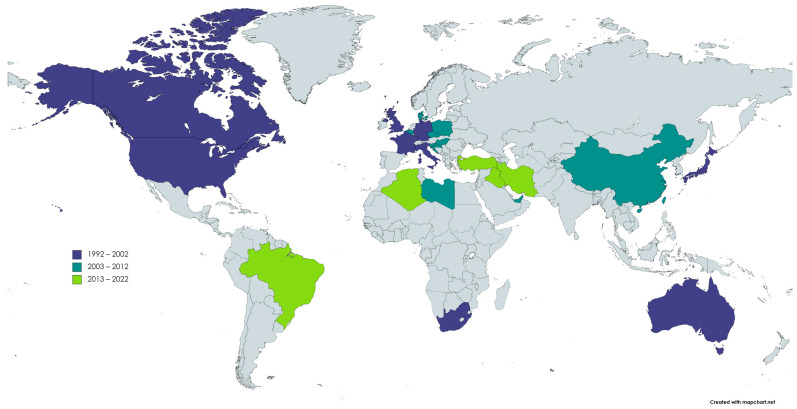
Countries with reported detection of PiCV colored according to the earliest retrieved record of detection report.

**Figure 6 viruses-14-01498-f006:**
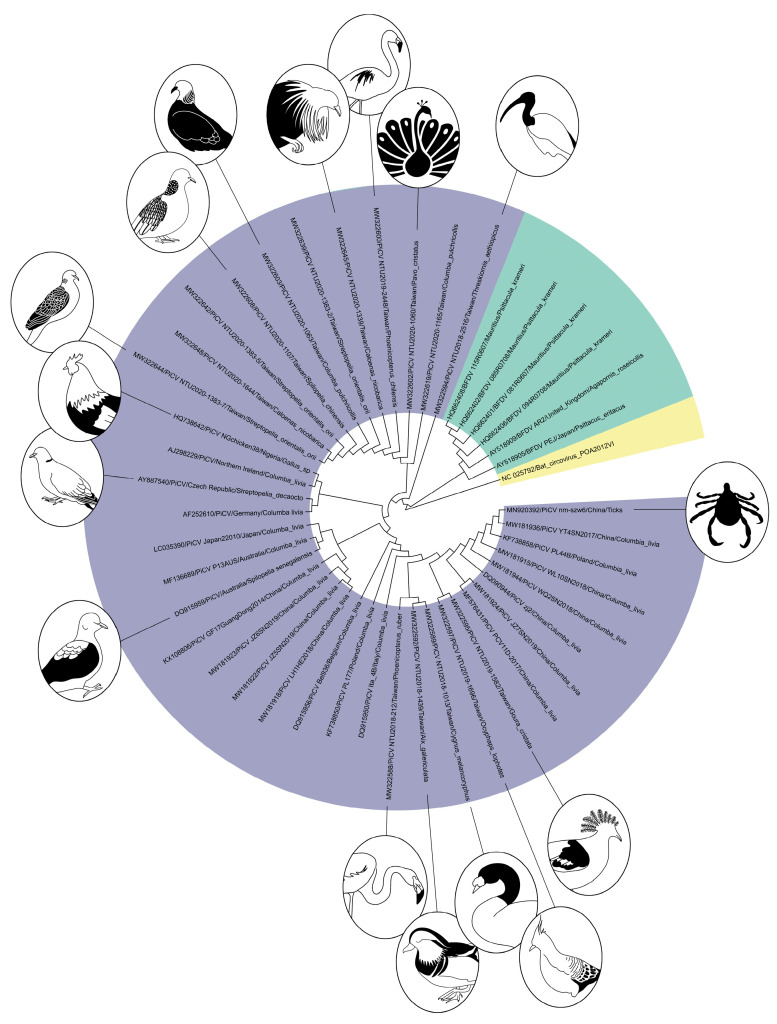
A neighbor-joining tree depicting the relationship among PiCV sequences from various hosts.

**Table 1 viruses-14-01498-t001:** Sources with the greatest number of records.

Sources	Count
Avian Pathology	15
Avian Diseases	8
Veterinary Record	8
Journal of Avian Medicine and Surgery	5
Archives of Virology	5
Medycyna Weterynaryjna	5
Journal of Veterinary Diagnostic Investigation	4
Veterinary Microbiology	4
Journal of General Virology	4
Virus Research	3
PLoS ONE	3

**Table 2 viruses-14-01498-t002:** Authors with the greatest number of records.

Author	Documents	Citations ^a^	Total Link Strength	Most Recent Record (Year)
Todd, D.	21	865	85	2012
Smyth, J.A.	12	460	57	2011
Duchatel, J.P.	11	169	54	2011
Stenzel, T.A. *	10	140	38	2021
Chuang, K.-P. *	6	26	39	2021
Koncicki, A. *	6	75	24	2017
Mankertz, A.	6	192	19	2011
Shivaprasad, H.L. *	5	134	33	2019
Tykalowski, B. *	5	59	23	2020
Muller, H.	5	200	22	2008
Raidal, S.R. *	5	127	18	2019
Soike, D.	5	304	15	2002

^a^ as indexed in the Dimensions database; * authors with new records within the last five years.

**Table 3 viruses-14-01498-t003:** Proposed terminology for pigeon circovirus diseases together with their case definition based on clinical and laboratorial findings.

PiCVD Proposed Name (Acronym)	Main Clinical Sign	Individual Diagnostic Criteria
PiCV-subclinical infection (PiCV-SI)	No evident clinical sign	Lack of overt clinical signs No or minimal histologic lesions in the lymphoid organs, mainly bursa of Fabricius and/or spleen Detection of PiCV at least from fecal or cloacal swab samples by standard or quantitative PCR
PiCV-systemic disease (PiCV-SD)	Lethargy, depression, weight loss, diarrhea, vomiting	Clinical signs Histologic lesions with characteristic intranuclear and/or intracytoplasmic viral inclusions mainly in the bursa and/or in the spleen Moderate to high amount of PiCV in damaged tissue, demonstrable by electron microscopy, ISH, and/or quantitative PCR

## Data Availability

Not applicable.

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
