# Peer review of "Pigeon Circovirus over Three Decades of Research: Bibliometrics, Scoping Review, and Perspectives"

_viruses, 2022, doi:10.3390/v14071498_

Round 1
Reviewer 1 Report
The review used bibliometric method to map the entire accessible PiCV-related research corpus with the aim of understanding its present research landscape, particularly in consideration of its historical context. In section 1 and 2, the authors described how to analyse literature by bibliometric method. However, the term of bibliometrics did not appear in the title of the manuscript. So, I suggest add "bibliometrics" into the title of the paper.
Author Response
The authors would like to express their gratitude to the reviewer for their favorable assessment of the manuscript.
The title of the paper was edited to reflect the suggestion of the reviewer. “Bibliometrics” was included in the title. Please see Line 3.
The new title of the paper now is:
Pigeon circovirus over three decades of research: Bibliometrics, Scoping Review, & Perspectives

Reviewer 2 Report
Dear Authors,
this review is an interesting approach to PiCV topic. The paper is well written and proposed terminology of PiCV infection is worth mentioning. I have no find any serious errors of the manuscript. However, in my opinion in the part of 3.1 subsection concerning the frequency of mutations and genetic diversity of the virus the phylogenetic tree based of all currently availble PiCV full genome sequences could be very useful here. The authors can consider creating two trees - one based on sequnces downloaded from the NCBI database, and the second one concerned sequences obtained after recombinant fragments of the genome removed.
Author Response
The authors would like to express their gratitude to the reviewer for their favorable assessment of the manuscript.
We would also like to thank the reviewer for their suggestion. In response to this, the sections on mutations and genetic recombination were moved to the end portion of 3.2 as this better fit the focus of this section, i.e., genetic diversity. We also considered providing the suggested trees, which were provided as supplementals as similar trees and/or similar analyses were already provided in other papers (particularly the whole genome trees in Khalifeh, 2021). Additional discussion about genetic recombination was also provided, with our notes on the possible weakness of the current attempt to come up with a PiCV genomes association based on a global dataset. We also emphasized the need for additional whole genome sequences to resolve this. See Lines 410-493.

Reviewer 3 Report
To Editor, Authors
No doubt, the Review “Pigeon circovirus over three decades of research: Review & Perspectives” is of interest for virologists. And also molecular epidemiologists of circoviruses and veterinarian authorities. Authors analyzed knowledge gaps and open research questions relevant to future PiCV-related studies. Proposed definitions and diagnostic criteria for PiCV subclinical infection (PiCV-SI) and PiCV systemic disease (PiCV-SD) were also provided. It contains 123 references.
Finally, this is excellent review for all who are interested in Pigeon circovirus. I really enjoyed this all. However, I don't feel qualified to judge about the English language and style. The manuscript can be accepted after checking of this.
Just comment:
How could you explain decrease number of new records in 2005 (Figure 2,b)?
Author Response
The authors would like to express their gratitude to the reviewer for their favorable assessment of the manuscript.
The other observation of the reviewer regarding the decrease of the number of new records in 2005 is indeed an interesting one. While we may not be able to actually find out the reason to this, we think that the 2001-2005 decline period is attributable to the relatively difficult method to assess infection of this virus, ie. electron microscopy, and molecular- or immuno-histopathology. We also note that in the same period, which followed the year when the first PiCV whole genome sequence was made available, PiCV studies were focused on the development and testing of new molecular methods for the detection of the virus that are more specific and/or easier to perform.
Lines 141-148 were added to reflect this thinking.
